# Efficacy of a Modified Live Porcine Reproductive and Respiratory Syndrome Virus 1 (PRRSV-1) Vaccine against Experimental Infection with PRRSV AUT15-33 in Weaned Piglets

**DOI:** 10.3390/vaccines10060934

**Published:** 2022-06-11

**Authors:** Sophie Duerlinger, Christian Knecht, Spencer Sawyer, Gyula Balka, Marianne Zaruba, Till Ruemenapf, Christian Kraft, Poul Henning Rathkjen, Andrea Ladinig

**Affiliations:** 1University Clinic for Swine, Department for Farm Animals and Veterinary Public Health, University of Veterinary Medicine, 1210 Vienna, Austria; christian.knecht@vetmeduni.ac.at (C.K.); spencer.sawyer@vetmeduni.ac.at (S.S.); andrea.ladinig@vetmeduni.ac.at (A.L.); 2Department of Pathology, University of Veterinary Medicine, 1078 Budapest, Hungary; balka.gyula@univet.hu; 3Institute of Virology, Department for Pathobiology, University of Veterinary Medicine, 1210 Vienna, Austria; marianne.zaruba@vetmeduni.ac.at (M.Z.); till.ruemenapf@vetmeduni.ac.at (T.R.); 4Boehringer Ingelheim Vetmedica GmbH, 55216 Ingelheim, Germany; christian.kraft@boehringer-ingelheim.com; 5Boehringer Ingelheim Animal Health Nordics, 2100 Copenhagen, Denmark; poul.rathkjen@boehringer-ingelheim.com

**Keywords:** porcine reproductive and respiratory syndrome virus (PRRSV), PRRSV-1 AUT15-33, respiratory model, weaned piglets, challenge model, modified live virus vaccine

## Abstract

In this study, the efficacy of the commercial modified live PRRSV-1 vaccine “Ingelvac PRRSFLEX^®^ EU” was assessed in weaned piglets experimentally infected with PRRSV strain AUT15-33. Seventy-four weaned piglets were allocated to five groups. Vaccinated (groups 1, 2, and 5) and non-vaccinated piglets (groups 3 and 4), infected with either a low dose (10^3^ TCID_50_/dose; groups 2 and 4) or a high dose (10^5^ TCID_50_/dose; groups 1 and 3) of the virus, were compared regarding clinical signs, average daily weight gain (ADG), lung lesions, viral load in serum, oral swabs, and tissue samples. In comparison to vaccinated animals, coughing increased notably in the second week after challenge in non-vaccinated piglets. During the same time period, vaccinated, high-dose-infected piglets showed significantly higher ADG (*p* < 0.05) than non-vaccinated, high-dose-infected animals. All infected piglets reached approximately the same viremia levels, but vaccinated animals showed both a significantly reduced viral load in oral fluid (*p* < 0.05) and tissue samples and significantly reduced lung lesions (*p* < 0.05). In conclusion, vaccination was able to increase ADG, reduce the amount of viral shedding via oral fluids, and reduce the severity of lung lesions and the viral load in tissue samples under experimental conditions.

## 1. Introduction

Porcine reproductive and respiratory syndrome virus (PRRSV) still has an important economic impact on pig production worldwide. The virus is responsible for losses in breeding as well as in growing pig herds [1]. PRRSV is a small enveloped positive-sense single-stranded RNA virus with a high genetic diversity and belongs to the virus family *Arteriviridae,* order *Nidovirales* [2,3]. There are two species defined [4,5], *Betaarterivirus suid 1* (PRRSV-1, previously European genotype 1) and *Betaarterivirus suid 2* (PRRSV-2, previously North American genotype 2). Additionally, there is significant genetic variability among different PRRSV isolates within each species [6,7,8].

PRRSV infection of pigs can occur via several routes, including intranasal, oral [9,10], intramuscular, intrauterine [11], or vaginal [12,13]. The major clinical signs of PRRS had already been described before the etiology of the disease was known: late-term abortions, mummified fetuses, stillborn or weak piglets, anorexia, fever, cyanosis, reduced growth rates, and post-weaning respiratory problems in piglets [14,15]. However, clinical presentation varies considerably between herds, from asymptomatic to devastating clinical signs, including high rates of mortality, and is influenced by virulence differences among PRRSV isolates, host immune status, host susceptibility, concurrent infections, and several other management factors [16].

In contrast to the reproductive disease in sows, the respiratory symptoms and lesions of PRRSV mono-infections in growing pigs are more difficult to reproduce under experimental conditions, where pigs live under optimal conditions in terms of space, temperature, and air quality. In contrast, infections in the field are often complicated by secondary infections, leading to massive financial losses [1,17].

Strategies to control or avoid PRRSV include strict management programs, such as biosecurity measures and vaccination [18]. Clinical signs caused by PRRSV are often managed using modified live virus (MLV) vaccines. They induce effective immune responses, including both humoral and cellular components [19]. One of the major objectives in the control of PRRSV with the use of MLV vaccines is the reduction in shedding and transmission of the virus [20].

PRRSV strain AUT15-33, which was used for the experimental infection of piglets in this study, was detected in an Austrian piglet-producing farm in 2015, where it caused a severe clinical outbreak [21]. 

The reproductive syndromes caused by this virus strain were already confirmed in an experimental infection of pregnant gilts [22]. Thereafter, PRRSV-1 strains closely related to AUT15-33 were found in different regions of Austria and in Germany [22]. In the current study, we wanted to assess whether AUT15-33 is also virulent in a respiratory model in weaned piglets. In the course of this study, the efficacy of the MLV vaccine “Ingelvac PRRSFLEX^®^ EU” (Boehringer Ingelheim Vetmedica GmbH, Ingelheim am Rhein, Germany) in decreasing clinical signs, improving weight gain, and reducing viral shedding, lung lesions, and viral loads in sera and tissue samples after experimental infection with PRRSV strain AUT15-33 was studied. In addition, two different challenge doses were used to determine which dose is capable of causing clinical symptoms and lesions in nursery piglets under experimental conditions.

## 2. Materials and Methods

### 2.1. Experimental Design

The entire experiment was carried out according to current Hungarian animal welfare regulations under the ethical permission number: BA02/2000-43/2017.

The study included seventy-four weaned piglets of PRRSV- and *Mycoplasma hyopneumoniae*-negative origin. The piglets were vaccinated against PCV-2 and tested negative for Swine Influenza Virus antibodies by ELISA (IDEXX Swine Influenza Virus Ab Test, IDEXX Montpellier SAS, Montpellier, France). Upon the arrival of the animals, the PRRSV-free status of the piglets was confirmed by measuring PRRSV-specific antibodies by ELISA (IDEXX PRRS X3 Ab Test, IDEXX Europe B.V., Hoofddorp, The Netherlands). They arrived at the biosafety level 3 facility at 3 weeks of age, and each animal was individually marked with a numbered ear tag. Piglets were divided into 5 groups: groups 1 to 4 included 16 animals each, while 10 animals were kept as controls in group 5. The groups were housed separate from each other, with piglets of groups 1 and 2 and piglets of groups 3 and 4 sharing the same air space (Table 1).

At four weeks of age (study day 0), animals of groups 1, 2, and 5 were vaccinated intramuscularly with a dose of 1 mL of the commercial MLV vaccine Ingelvac PRRSFLEX^®^ EU (Boehringer Ingelheim Vetmedica GmbH, Germany), a live attenuated PRRSV-1 vaccine containing strain 94881 (full genome Gen Bank accession number KT988004) at a minimum dose of 10^4.4^—10^6.6^ TCID_50_ (Tissue Culture Infectious Dose 50). The animals of groups 3 and 4 were sham-treated intramuscularly with 1 mL of saline solution per piglet on study day 0.

On study day 28, piglets of groups 1 to 4 were inoculated intranasally with the PRRSV isolate AUT15-33. They were retained by use of a snare, and the virus was vaporized directly into the nostrils of the piglets within seconds (5 mL of cell culture supernatant including the virus per side) using a mucosal atomization device (LMA MAD Nasal™, Teleflex Medical GmbH, Athlone, Ireland). The particle size of 30–100 µm enables the rapid absorption of the inoculum via the mucous membranes. Two different infection doses were used: 10^5^ TCID_50_ for piglets in groups 1 and 3, and 10^3^ TCID_50_ for piglets in groups 2 and 4. The animals of group 5 were sham-inoculated intranasally with cell culture medium (the same medium as used for the culture of PAMs to grow the virus) on study day 28.

Any cough that occurred was monitored and recorded from study days 0 to 42 using the “cough monitor” (Pig respiratory distress package, SoundTalks NV, Belgium). The package contains a portable hardware platform (SOMO) to process audio in real time, the SoundTalks microphone, and the pig respiratory distress monitoring software. One microphone was placed in each room (room A: both vaccinated, infected groups (groups 1 and 2); room B: both non-vaccinated, infected groups (groups 3 and 4); room C: vaccinated control group (group 5)). The output of this cough monitoring is an algorithm-based respiratory distress index that takes the number of coughs and the number of pigs in the room into account.

Clinical observation, including parameters such as behavior, appetite, dyspnea, coughing, and nasal or eye discharge, was performed once a day from study day 28 to study day 70 by a blinded observer. The same person performed the observations throughout the entire study period. The measurement of rectal temperature was performed for the first 14 days post-challenge, from study day 28 to study day 42. The body weight of the animals was recorded on study days 0, 28, 35, 42, and 70 to calculate the mean of the average daily weight gain (ADG) for each group for the time period prior to challenge (from study day 0 to 28), from days 28 to 35 (first week post-challenge), from days 35 to 42 (second week post-challenge), and from days 42 to 70 (first necropsy until necropsy of the remaining animals).

Half of the animals from each treatment group were euthanized and necropsied on study day 42. The remaining animals were followed until study day 70, when necropsies were performed (Figure 1).

### 2.2. Challenge Virus

The PRRSV-1 isolate AUT15-33 (Gen Bank accession number MT000052.1) was used as the challenge virus. The virus was isolated for the first time in 2015 from sera of piglets showing acute illness by passage on porcine alveolar macrophages (PAMs), partially sequenced (ORF2-7) and grouped as PRRSV-1, subtype 1 [21]. Kreutzmann et al. compared AUT15-33 to different PRRSV strains, of which the phenotypic characterization and whole-genome sequence have already been published [22]. For the current study, the virus was propagated on primary porcine alveolar macrophages for three passages to obtain 50 mL of virus stock (5.6 × 10^5^ TCID_50_/mL) for challenge infection. PAMs were prepared as described in [21] by lavage of the lung of a euthanized 10-week-old pig. Titrations were carried out on PAMs using indirect immunofluorescence with a monoclonal antibody anti-N [23].

### 2.3. Sample Collection and Necropsy

Blood samples, as well as oral swabs (sterile dry cotton swabs), were collected on study days 0, 14, 21, 28, 31, 33, 35, 37, 39, 42, 49, 56, 63, and 70 to evaluate viral RNA copy numbers in sera and in swabs. Serum was also used to detect PRRSV-specific antibodies. The blood samples were centrifuged, and the sera were collected in capped tubes and frozen at –80 °C. Oral swabs were also stored at –80 °C until further processing.

At termination, every piglet was humanely euthanized according to current animal welfare regulations in Hungary. Complete necropsies were performed on all piglets, and pathologic findings were recorded. Tissue samples were collected to assess viral loads in the lungs, tracheobronchial lymph nodes, and tonsils. One sample of each of the seven lung lobes (2 × 2 cm) of each piglet was fixed in 10% neutral buffered formaldehyde solution for 24–48 h, dehydrated with a series of alcohol and xylene, and subsequently embedded in paraffin wax. Then, 3–4 µm thick slices were cut and mounted on glass slides for routine staining with hematoxylin and eosin for histologic evaluation of lung lesions.

Gross lung lesions, such as tan mottled areas and areas of firm consistency of each lung lobe, were macroscopically classified according to the percentage of the affected lung lobe. The relative weights of the different pulmonary lobes according to Christensen et al. (1999) were used to calculate the total weighted lung lesion score [24].

### 2.4. Histologic Lung Assessment

One blinded pathologist scored seven lung lobes from each piglet histologically for severity and extension (severity scored as 0: no lesion; 1: mild; 2: moderate or 3: severe, extension: 0: not present; 1: focal; 2: multifocal; 3: diffuse distribution of the given lesion) of the following five lesions: pneumocytic hypertrophy and hyperplasia, septal infiltration with mononuclear cells, intra-alveolar necrotic debris, intra-alveolar inflammatory cell accumulation, and perivascular inflammatory cell accumulation. The total histologic score (histo score) was calculated from the sum of the lesion severity and lesion extension of each histologically examined parameter in all lung lobes (maximum score of 42 per lesion per piglet; maximum total score of 210 per piglet) [25]. Hematoxylin-and-eosin-stained slides were scanned and digitalized with the Pannoramic Midi slide scanner (3D Histech, Budapest, Hungary). The representative images were obtained with the SlideViewer software (3D Histech).

### 2.5. Serum Antibody Detection

A commercial enzyme-linked immunosorbent assay (ELISA) kit (IDEXX PRRS X3 Ab Test, IDEXX Europe B.V., Hoofddorp, The Netherlands) was used to detect PRRSV-specific antibodies in serum samples collected on study days 0, 14, 28, 35, 42, and 70. The ELISA was performed according to the manufacturer’s instructions; samples with a sample-to-positive control (S/P) ratio equal to or higher than 0.4 were considered to be positive for PRRSV-specific antibodies.

### 2.6. Virological Analysis—PRRSV qRT-PCR

Viral loads were determined as described by Kreutzmann et al. [22]. In brief, AUT15-33-optimized ORF7-based qPCR was employed using primers and probes as used by Kreutzmann et al. [22]. The only difference was the qPCR platform, as a Rotorgene 5plex (Qiagen GmbH, Hilden, Germany) has replaced the Perkin Elmer 7300 instrument used by Kreutzmann et al.

Tissue and organ sections (50 mg) were extracted with 600 µL of Qiazol (Qiagen GmbH, Hilden, Germany) using 3 stainless steel beads (3 mm) in a 2 mL screwcap tube (SARSTEDT AG & Co. KG, Nümbrecht, Germany) for homogenization in a TissueLyser II instrument (Qiagen GmbH, Hilden, Germany) at full speed for 3 min. The homogenate was briefly centrifuged, and 300 µL of chloroform was added. The capped tubes were thoroughly vortexed and centrifuged for phase separation at 13,000× *g* for 5 min. Then, 200 µL of the aqueous phase was collected and further processed using the ViralPathogen Kit in a QiaCubeHT instrument (Qiagen GmbH, Hilden, Germany) according to the manufacturer’s protocol. Two microliters of the eluted RNA was used for ORF7-specific RT-qPCR using the Luna Onestep RT PCR Kit (New England Biolabs). The primer sequences were adapted from Egli et al. [26] to fit the sequence of PRRSV-1 strain AUT15-33 (PRSq1 forward: TCAACTGTGCCAGTTGCTGG; PRSq2 reverse: TGRGGCTTCTCAGGCTTTTC; and PRSq3 probe: 5′Fam-CCCAGCGYCRRCARCCTAGGG Tamra-3′). To assess the presence of the vaccine strain contained in Ingelvac PRRSFLEX^®^ EU, the primer set PRSq1 forward: TCAACTGTGCCAGTTGCTGG, PRSq4 reverse: TGTGGCTTCTCAGGCTTCTTC and PRSq5 probe: 5′Fam CCCAGCGCCAGCAAYCTAGGG Tamra-3′ was employed. The detection of β-actin was performed as described by Kreutzmann et al. [22] as the extraction control.

### 2.7. Statistical Analysis

Statistical analysis was performed with IBM SPSS^®^ Statistics (version 25.0). All collected data were tested for normal distribution using the Shapiro–Wilk test. The main objective was to compare vaccinated with non-vaccinated piglets infected with either a high dose or low dose of PRRSV in order to evaluate vaccine efficacy. For this comparison, T-tests with Bonferroni post hoc corrections were used for data showing normal distributions. Rectal body temperature was compared among vaccinated and non-vaccinated piglets infected with either a high dose or low dose of PRRSV using repeated measures ANOVA with Bonferroni post hoc corrections. For data without a normal distribution, non-parametric tests (Kruskal–Wallis test/Mann–Whitney U test) were used to compare the results between the respective groups; *p*-values < 0.05 were considered statistically significant, and *p*-values < 0.1 were considered numerical differences.

## 3. Results

### 3.1. Clinical Observation

No piglets died due to PRRSV AUT15-33 infection in the current study. During the daily clinical examination, coughing and dyspnea were observed occasionally in the infected groups, both vaccinated and non-vaccinated. The rectal body temperature of the vaccinated, infected piglets showed no statistically significant difference to the body temperature of the non-vaccinated, infected piglets, neither between low-dose-infected groups nor between high-dose-infected groups (Figure 2). No further clinical signs were observed during the daily examination.

### 3.2. Cough Monitor

The cough monitor, which collected all cough events occurring from study day 0 to study day 42, recorded more coughing than observed during the clinical examination. There was a notable increase in cough events in the non-vaccinated, infected groups in the second week after challenge, but no statistically significant differences in the area under the curve (AUC) of cough events were detected between vaccinated and non-vaccinated, infected piglets (Figure 3).

### 3.3. Average Daily Weight Gain

The average weight of the piglets per treatment group on study day 0 was between 6.5 and 7.5 kg, and the differences between treatment groups were not statistically significant. The average daily weight gain of each treatment group for different time periods is shown in Figure 4. In the first time period (prior to challenge, study days 0–28), the mean ADG of the piglets ranged from 427.9 g to 542.9 g without significant differences between vaccinated and non-vaccinated groups (*p* > 0.05). In the second week after challenge (study days 35–42), vaccinated, high-dose-infected piglets showed significantly higher ADG (*p* < 0.05) than non-vaccinated, high-dose-infected animals. During the same time period, vaccinated, low-dose-infected piglets showed a numerically higher ADG (*p* < 0.1) compared to non-vaccinated, low-dose-infected piglets. In the last time period (study days 42–70), vaccinated, low-dose-infected piglets again showed a numerically higher ADG (*p* < 0.1) compared to non-vaccinated, low-dose-infected piglets.

### 3.4. Serology

PRRSV-specific antibodies in serum were detected on study day 14 in all vaccinated animals, while non-vaccinated animals remained negative in the time period prior to challenge (Figure 5).

After infection on study day 28, the non-vaccinated, high-dose-infected animals showed a faster increase in the S/P ratio (mean ± sd S/P ratio on study day 35: 0.12 ± 0.08; study day 42: 1.6 ± 0.29) compared to the animals of the non-vaccinated, low-dose-infected group (study day 35: 0 ± 0.03; study day 42: 1.04 ± 0.56).

### 3.5. Viral Load in Serum

All animals were monitored for the presence of MLV with PRRSFLEX^®^ EU–specific qPCR on study days 0, 14, 21, 28, 31, 33, 35, 37, 39, 42, 49, 56, 63, and 70. Viral genomes were detected on study day 14 in two, on day 21 in three, on day 28 in six, and on day 31 in three vaccinated animals. After study day 31, the vaccine virus was no longer detected by PRRSFLEX^®^ EU–specific qPCR.

The viral load in serum, which was determined using AUT15-33-specific qRT-PCR, which is not cross-reactive with the MLV genome, increased in all infected piglets after challenge. No significant differences were detected in the AUC of viral load in the serum between the infected groups. Non-infected animals remained negative in AUT15-33-specific PCR. Viremia was first detected in all groups of infected animals on study day 31, 3 days post-infection (dpi). A slower increase in viral load in sera was measured in the groups that were infected with the lower dose than in those infected with the higher dose of the virus. The viral load of piglets in the respective non-vaccinated group increased more rapidly compared to piglets in the vaccinated group. On study day 39 (11 dpi), all infected piglets reached approximately the same viremia levels (average 10^8^ GE/mL). Thereafter, the viral load decreased in all four groups of infected animals. By study day 70, only a few animals remained positive in AUT15-33-specific qRT-PCR (Figure 6).

### 3.6. Viral Load in Oral Swabs

All infected animals shed the virus through oral fluids. Piglets from the vaccinated, infected groups shed less virus than the non-vaccinated, infected animals, and the duration of shedding was shorter in vaccinated piglets than in non-vaccinated piglets (Figure 7). A significant difference (*p* < 0.05) was detected in the area under the curve (AUC) of viral shedding through oral fluids between the vaccinated, low-dose-infected piglets and the non-vaccinated, low-dose-infected piglets, as well as between the vaccinated, high-dose-infected and the non-vaccinated, high-dose-infected piglets.

Piglets that were infected with the lower PRRSV dose (vaccinated and non-vaccinated piglets) showed a rebound of shedding on study day 63. Afterwards, the amount of shed virus decreased in both groups. On study day 70, the viral load in oral swabs was lower than on day 63 in all groups except in the non-vaccinated, high-dose-infected piglets.

### 3.7. Macroscopic Lung Lesions

Animals from all infected groups showed gross lung lesions, such as tan mottled areas and areas of firm consistency (Figure 8). During the first necropsy (study day 42), macroscopic lung lesions were more distinct than during the second necropsy (study day 70). A significant difference (*p* < 0.05) in gross lung lesions between the vaccinated, high-dose-infected and the non-vaccinated, high-dose-infected piglets was detected, while there was a numerical difference (*p* < 0.1) between vaccinated, low-dose-infected and non-vaccinated, low-dose-infected animals during the first necropsy. During the second necropsy, no more significant differences in macroscopic lung lesions were found.

### 3.8. Histologic Lung Lesions

Histologic lung lesions were found in animals of all groups (Figure 9). Regarding the five different histologic lesions that were examined (pneumocytic hypertrophy and hyperplasia, septal infiltration with mononuclear cells, intra-alveolar necrotic debris, intra-alveolar inflammatory cell accumulation, and perivascular inflammatory cell accumulation), significant differences (*p* < 0.05) were found in all lesions between vaccinated and non-vaccinated animals during the first necropsy (Figure 9A–E). During the first necropsy (study day 42), the total histo score was significantly higher in non-vaccinated compared to vaccinated animals (Figure 10). During the second necropsy, no significant differences were present between treatment groups, and lesions were less severe compared to the first necropsy time point (Figure 10 and Figure 11).

### 3.9. Viral Load in Tissue Samples

The viral load in lung tissue differed significantly (*p* < 0.05) between vaccinated and non-vaccinated infected animals during the first necropsy. In lung tissues from the second necropsy, no more statistically significant differences were present in viral levels between animals of vaccinated and non-vaccinated groups (Figure 12A).

Viral loads in tracheobronchial lymph nodes differed significantly (*p* < 0.05) between vaccinated, high-dose-infected and non-vaccinated, high-dose-infected animals during the first necropsy. Viral levels in tracheobronchial lymph nodes during the second necropsy differed significantly between vaccinated, high-dose-infected and non-vaccinated high-dose-infected animals, as well as between vaccinated, low-dose-infected and non-vaccinated, low-dose-infected animals (Figure 12B).

Viral loads in tonsils differed significantly (*p* < 0.05) between vaccinated, low-dose-infected and non-vaccinated, low-dose-infected animals during the first necropsy. Otherwise, no significant differences in viral loads of tonsil tissue were detected between the vaccinated and non-vaccinated animals (Figure 12C).

## 4. Discussion

Although it is difficult to experimentally reproduce respiratory symptoms as a consequence of PRRSV mono-infection, particularly using PRRSV-1 field isolates [27,28,29], we managed to provoke clinical signs and lung lesions in weaned piglets after experimental infection with PRRSV strain AUT15-33, both with the low dose (10^3^ TCID_50_/dose) and with the high dose (10^5^ TCID_50_/dose) of the virus. Since the experimental infection was successful in the current study, the efficacy of a PRRS-MLV vaccine could be assessed by comparing clinical signs, viral load in serum, viral load in oral swabs (viral shedding), macroscopic and histologic lung lesions, and viral load in tissue samples of vaccinated and non-vaccinated animals after PRRSV infection. The investigated parameters were used to evaluate vaccine efficacy, since viremia levels alone are inadequate [30,31].

A weakness of the current study was the fact that animals of the low-dose-infected groups were housed in the same airspace as the animals of the respective high-dose-infected group. However, vaccinated and non-vaccinated pigs were housed in different air spaces, and the piglets of the groups housed in the same air space did not have direct contact with piglets of the respective other group. In addition, separate equipment, clothing, boots, and gloves were used for the different groups. The initial difference and subsequent accordance of the viral load in sera between low-dose- and high-dose-infected animals suggest that infection with different doses worked out despite the suboptimal housing conditions.

In the current study, hardly any coughing was observed during the daily clinical examination, while the cough monitor recorded more coughing and a notable increase in cough events in non-vaccinated, infected animals in the second week after infection. These results show that the use of the cough monitor enables an objective, continuous measurement of respiratory symptoms compared to the very subjective clinical examination by humans once a day. In addition to the increased cough events, there was a significantly (*p* < 0.05) lower average daily weight gain in non-vaccinated, high-dose-infected piglets compared to vaccinated, high-dose-infected animals in the same time period. For the average daily weight gain, it must be kept in mind that there were no field conditions in this study. However, several studies performed under field conditions also showed a higher average daily weight gain in MLV-vaccinated compared to non-vaccinated pigs [32,33,34]. Due to the fact that about 55% of the costs of PRRSV infection in general in the US are caused by infection of growing pigs [1], vaccination of piglets can be a useful strategy to reduce clinical signs and thus costs [35,36].

ELISA S/P ratios are not related to protection from PRRSV infection [37]; rather, they show whether the animals have had prior contact with PRRSV, either by vaccination with MLV vaccine or by field virus infection. In the current study, PRRSV-specific antibodies were detected in all vaccinated animals after vaccination and prior to PRRSV challenge. In the non-vaccinated, infected animals, antibodies were detected only after infection, suggesting that none of the animals had prior contact with PRRSV before vaccination or infection. Viremia was not mitigated by vaccination in the current study. The viral load in serum increased in all infected piglets after challenge. The highest peak of viral load is typically reached 7–14 days post-PRRSV infection [16]. In this case, a slower increase in viral load in sera was measured in the groups that were infected with the lower dose than in those infected with the higher dose of the virus. Nevertheless, on study day 39 (11 dpi), all infected piglets reached approximately the same viremia levels. This initial difference and subsequent accordance of the viral load between low-dose- and high-dose-infected animals suggest that the virus isolate replicated very quickly, regardless of the infection dose. It has been described that a higher viral load in serum is detected after infection with highly virulent PRRSV strains, that the virus replicates to higher titers in younger pigs, and that the duration of viremia is prolonged in younger compared to older experimentally infected pigs [38,39,40,41].

In contrast to the results of the current study, Kreutzmann et al. described a significantly lower AUC of the serum viral load in vaccinated compared to non-vaccinated gilts, which were also infected with PRRSV-1 AUT15-33. The gilts were vaccinated with the same PRRSV-1 vaccine strain (94881, ReproCyc^®^ PRRS EU, Boehringer Ingelheim Vetmedica GmbH, Germany) as the piglets in the current study. However, the gilts were not vaccinated once, but twice before insemination and once in mid-gestation [22]. The time period between vaccination and infection is another factor influencing the effect of vaccination, as shown by Balka et al. (2016) in a field study in which piglets were vaccinated with “Ingelvac PRRSFLEX^®^ EU” 6–8 weeks prior to PRRSV infection and were thus protected from natural infection in terms of both viremia levels and the proportion of viremic animals [42]. Another difference in the study design between the recent study of Kreutzmann et al. and the current study was the challenge dose and the route of infection. The gilts in the above-mentioned study were inoculated both intramuscularly and intranasally with a total dose of 3 × 10^5^ TCID_50_/gilt [22]. The non-vaccinated gilts reached approximately the same viremia levels as the piglets in the current study, which is in contrast to the literature describing a higher viral load in younger pigs [39,43]. The relatively high viral load in sera, regardless of the age of the animals, and the appearance of clinical signs after experimental infection in both studies indicate that the PRRSV isolate AUT15-33 is of significant virulence. However, the route of infection must not be disregarded, as the various routes of exposure differ in the likelihood that a given dose will result in infection, and it is known that pigs are more susceptible to infection via parenteral exposure [44].

Although vaccination was not able to prevent viremia, it led to significantly lower excretion of the virus via oral fluids in vaccinated compared to non-vaccinated animals. This is one of the greatest benefits of vaccination, especially in areas with high pig density, since it reduces the probability of virus transmission from vaccinated animals to other animals [45]. This includes both virus transmission between pigs within a herd but also virus introduction into neighboring herds.

The two different necropsy time points in this study were chosen to assess macroscopic and histologic lung lesions and viral load in tissue samples at 14 dpi and 42 dpi. As expected and similarly seen in other studies [25], both macroscopic and histologic lung lesions were more severe at 14 dpi than at 42 dpi. Without secondary infections, the extent of lung involvement is noticeably decreased by 4 weeks after initial PRRSV exposure [46]. After the acute phase of PRRSV infection and in the absence of secondary bacterial pathogens, necrotic cells are removed, neutrophils are absent, and damaged lung tissue is replaced by proliferating type II pneumocytes [25]. Another interesting observation, especially at the first necropsy time point, was that low-dose-infected animals had a higher total lung lesion score and total histo score than high-dose-infected animals. This may be due to the chosen necropsy time point and the dose of infection. The high-dose-infected animals have probably already overcome the acute phase of the disease, and the lungs are already recovering. In the low-dose-infected piglets, the peak of viral replication might be delayed by a few days, which was also supported by the viremia data; additionally, more replicating virus was present in the lungs of low-dose-infected piglets at 14 dpi, explaining the higher lung lesion score.

In a previous study, Kreutzmann et al. found the lowest detection rate and viral load in the lungs compared to the tracheobronchial lymph nodes and tonsils of infected gilts around three weeks after infection. These findings may be related to the time point of necropsy, which was performed 3 weeks after experimental infection, and like the results of the current study, reflect the typical course of a PRRSV infection [22].

In the tracheobronchial lymph nodes, there was a significant difference (*p* < 0.05) in viral load between vaccinated, high-dose-infected and non-vaccinated, high-dose-infected animals not only during the first necropsy but also during the second necropsy in the current study. Kreutzmann et al. also detected a significantly lower viral load in tracheobronchial lymph nodes of vaccinated compared to non-vaccinated gilts three weeks after experimental infection with PRRSV AUT15-33 [22].

In the acute stage of infection, the lung is the preferential location for viral replication since porcine alveolar macrophages are the primary target cells for PRRSV [37,47]. During later stages of infection, viral replication is primarily localized in lymphoid organs such as the tonsils and lymph nodes, where the virus can persist for months [48,49]. The observed significantly lower viral load in tracheobronchial lymph nodes of vaccinated animals compared to non-vaccinated animals at both necropsy time points showed another positive effect of vaccination, as continuous viral replication in the lymph nodes contributes to the efficient transmission of the virus.

## 5. Conclusions

In conclusion, vaccination of piglets with the commercial MLV vaccine “Ingelvac PRRSFLEX^®^ EU” had a positive effect on average daily weight gain, reduced the amount and duration of viral shedding via oral fluids, reduced the severity of lung lesions, and led to a lower viral load in tissue samples after experimental infection with PRRSV AUT15-33 in weaned piglets. Even at low infectious doses, AUT15-33 provoked clinical signs and lung lesions in weaned piglets after experimental infection, thus confirming that this strain is a suitable challenge model for both the reproductive syndrome in sows and the respiratory disease in piglets.

## Figures and Tables

**Figure 1 vaccines-10-00934-f001:**
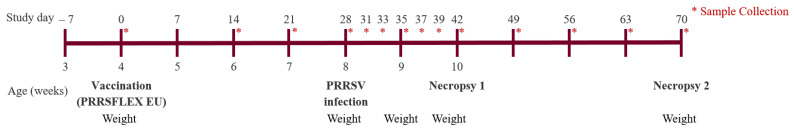
Timeline of the experiment. Study days on the top, age of the animals in weeks below.

**Figure 2 vaccines-10-00934-f002:**
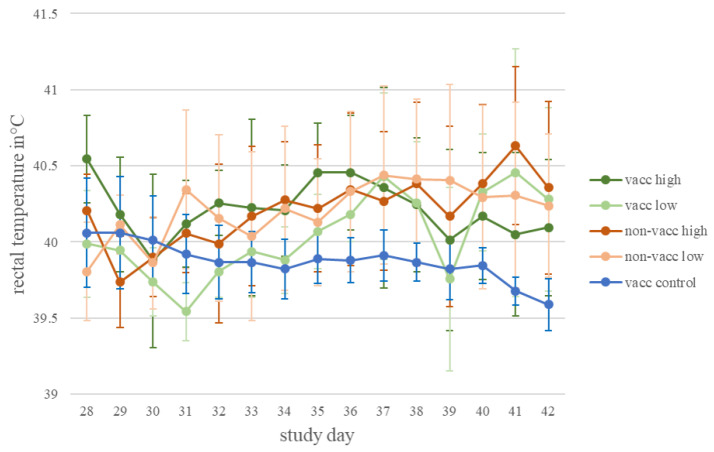
Rectal temperature. Mean and standard deviation of rectal temperature for each treatment group (vacc high: vaccinated, high dose [10^5^ TCID_50_/dose] infected; vacc low: vaccinated, low dose [10^3^ TCID_50_/dose] infected; non-vacc high: non-vaccinated, high dose infected; non-vacc low: non-vaccinated, low dose infected; vacc control: vaccinated, non-infected) after infection (study day 28 to 42).

**Figure 3 vaccines-10-00934-f003:**
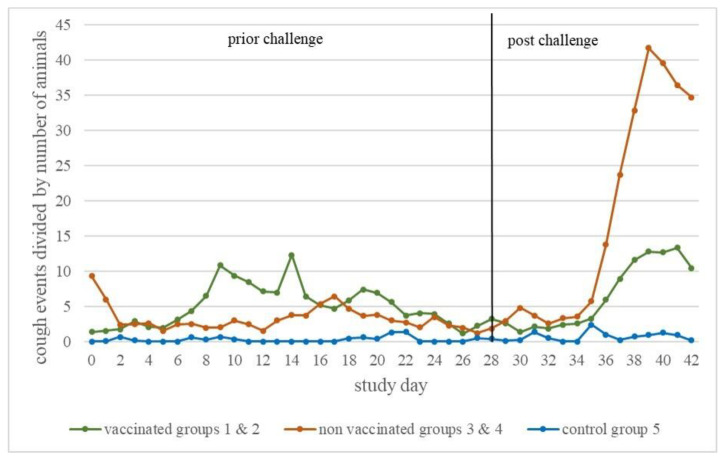
Cough monitor results. Cough events divided by the number of animals in each room (vaccinated, infected (groups 1 and 2), non-vaccinated, infected (groups 3 and 4), and vaccinated control (group 5) animals) per study day from study day 0 to study day 42.

**Figure 4 vaccines-10-00934-f004:**
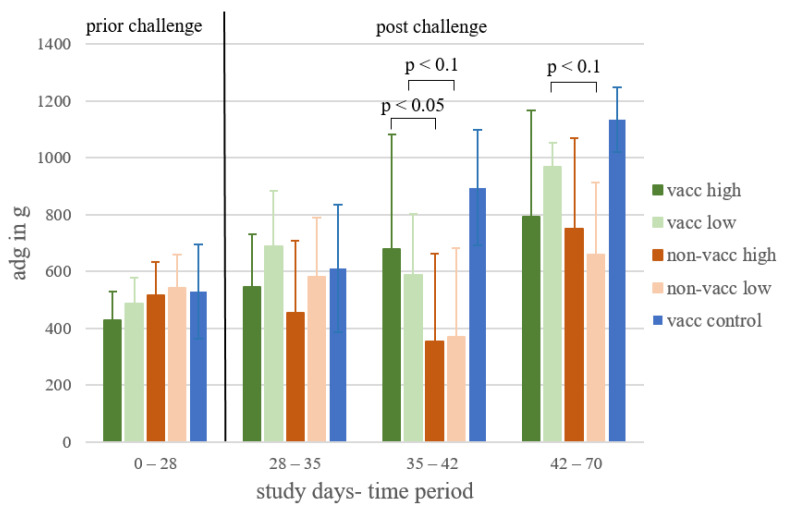
Average daily weight gain. Mean and standard deviation of average daily weight gain (ADG) for each treatment group (vacc high: vaccinated, high dose [10^5^ TCID_50_ /dose] infected; vacc low: vaccinated, low dose [10^3^ TCID_50_/dose] infected; non-vacc high: non-vaccinated, high dose infected; non-vacc low: non-vaccinated, low dose infected; vacc control: vaccinated, non-infected) for the different time periods: prior to infection, the first and second weeks after infection, and from two weeks after infection until termination. *p*-values < 0.05 and <0.1 are indicated using brackets.

**Figure 5 vaccines-10-00934-f005:**
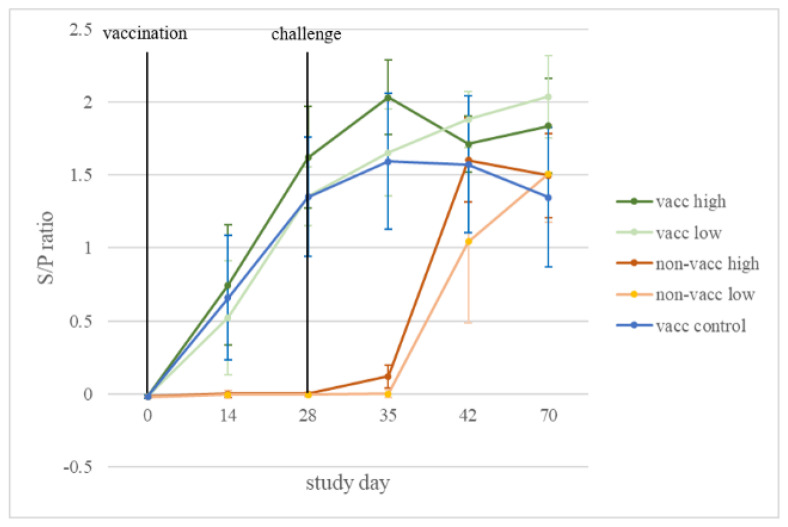
ELISA results. Mean and standard deviation of ELISA S/P ratios for each treatment group (vacc high: vaccinated, high dose [10^5^ TCID_50_ /dose] infected; vacc low: vaccinated, low dose [10^3^ TCID_50_/dose] infected; non-vacc high: non-vaccinated, high dose infected; non-vacc low: non-vaccinated, low dose infected; vacc control: vaccinated, non-infected) in serum samples collected over time.

**Figure 6 vaccines-10-00934-f006:**
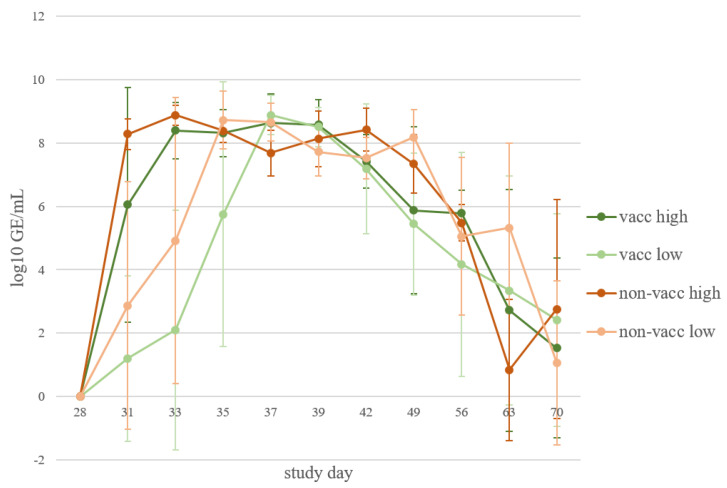
Viral load in serum. Mean and standard deviation of qRT-PCR results from serum samples (log_10_ GE/mL) of each infected group (vacc high: vaccinated, high dose [10^5^ TCID_50_ /dose] infected; vacc low: vaccinated, low dose [10^3^ TCID_50_/dose] infected; non-vacc high: non-vaccinated, high dose infected; non-vacc low: non-vaccinated, low dose infected) at different time points after infection (study days).

**Figure 7 vaccines-10-00934-f007:**
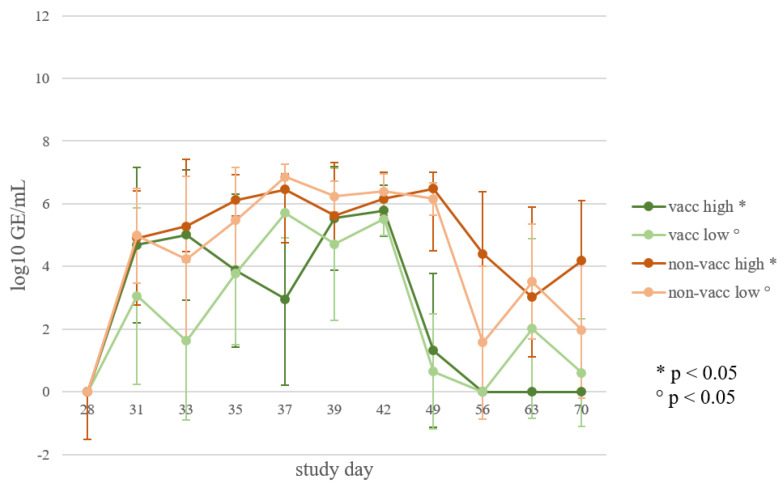
Viral load in oral swabs. Mean and standard deviation of qRT-PCR results from oral swabs (log_10_ GE/mL) of each infected group (vacc high: vaccinated, high dose [10^5^ TCID_50_ /dose] infected; vacc low: vaccinated, low dose [10^3^ TCID_50_/dose] infected; non-vacc high: non-vaccinated, high dose infected; non-vacc low: non-vaccinated, low dose infected) at different time points after infection (study days). Significant differences in the AUC values are indicated (*p* < 0.05).

**Figure 8 vaccines-10-00934-f008:**
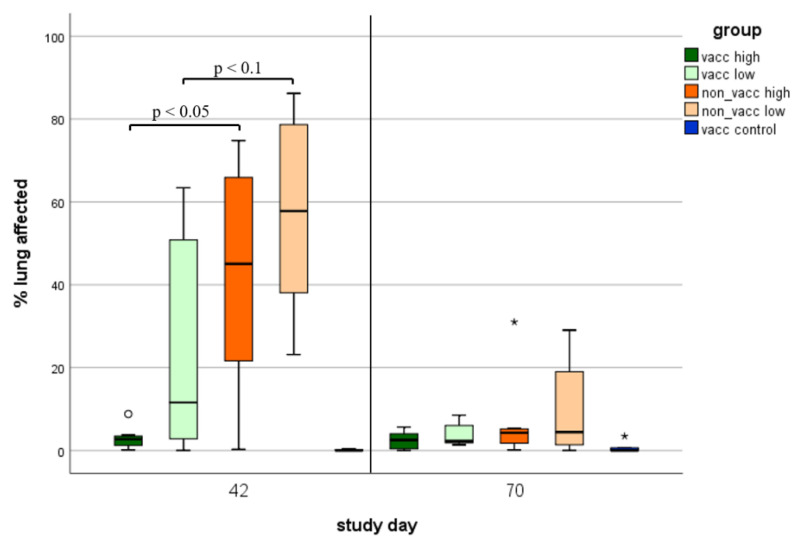
Macroscopic lung lesions. Total lung lesion score (corresponding to % of lung affected) of animals necropsied on either study day 42 (first necropsy) or study day 70 (second necropsy) in each group (vacc high: vaccinated, high dose [10^5^ TCID_50_ /dose] infected; vacc low: vaccinated, low dose [10^3^ TCID_50_/dose] infected; non-vacc high: non-vaccinated, high dose infected; non-vacc low: non-vaccinated, low dose infected; vacc control: vaccinated, non-infected). Boxplots show 25th and 75th percentiles and median values, and whiskers show minimum and maximum values within 1.5 × interquartile range (IQR). Outliers are values with more than 1.5 × IQR (indicated using circles °), and extreme values are values with more than 3 × IQR (indicated using asterisks *). *P*-values < 0.1 are indicated using brackets.

**Figure 9 vaccines-10-00934-f009:**
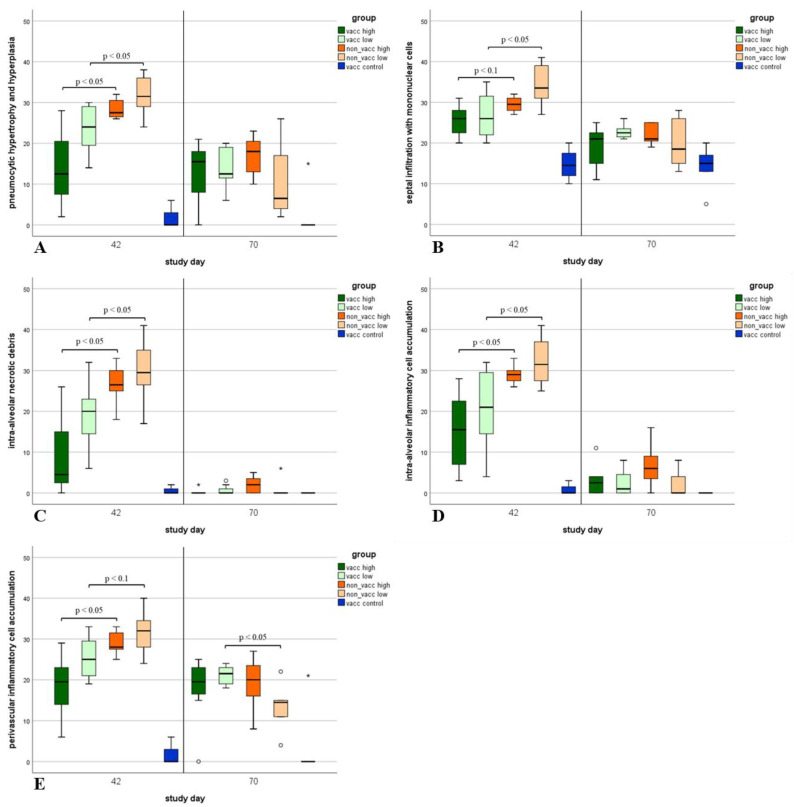
Histologic lung lesions. Histologic lung lesions of animals necropsied on either study day 42 (first necropsy) or study day 70 (second necropsy) in each group (vacc high: vaccinated, high dose [10^5^ TCID_50_ /dose] infected; vacc low: vaccinated, low dose [10^3^ TCID_50_/dose] infected; non-vacc high: non-vaccinated, high dose infected; non-vacc low: non-vaccinated, low dose infected; vacc control: vaccinated, non-infected). All seven lung lobes from each piglet were scored histologically for severity and extension (scored as 0: no lesion; 1: mild; 2: moderate; or 3: severe) of the following five lesions: pneumocytic hypertrophy and hyperplasia (**A**), septal infiltration with mononuclear cells (**B**), intra-alveolar necrotic debris (**C**), intra-alveolar inflammatory cell accumulation (**D**), and perivascular inflammatory cell accumulation (**E**). Maximum score was 42 per lesion per piglet. Boxplots show 25th and 75th percentiles and median values, and whiskers show minimum and maximum values within 1.5 × interquartile range (IQR). Outliers are values with more than 1.5 × IQR (indicated using circles °), and extreme values are values with more than 3 × IQR (indicated using asterisks *). *p*-values < 0.1 are indicated using brackets.

**Figure 10 vaccines-10-00934-f010:**
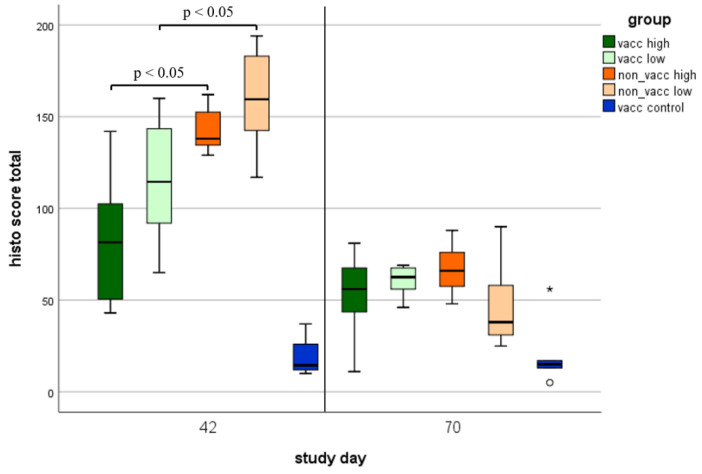
Total histo score. Total histo score of animals necropsied on either study day 42 (first necropsy) or study day 70 (second necropsy) in each treatment group (vacc high: vaccinated, high dose [10^5^ TCID_50_ /dose] infected; vacc low: vaccinated, low dose [10^3^ TCID_50_/dose] infected; non-vacc high: non-vaccinated, high dose infected; non-vacc low: non-vaccinated, low dose infected; vacc control: vaccinated, non-infected). Boxplots show 25th and 75th percentiles and median values, and whiskers show minimum and maximum values within 1.5 × interquartile range (IQR). Outliers are values with more than 1.5 × IQR (indicated using circles °), Extreme values are values with more than 3 × IQR (indicated using asterisks *). Significant differences (*p* < 0.05) between vaccinated and non-vaccinated piglets are indicated.

**Figure 11 vaccines-10-00934-f011:**
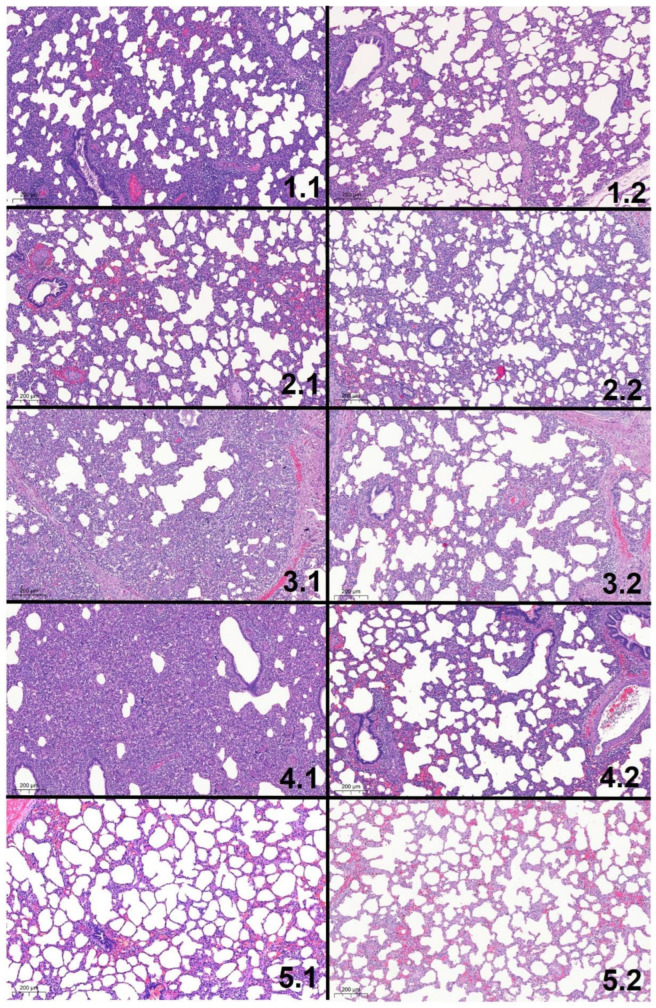
Representative histologic pictures of lung tissues obtained from the different groups at both necropsy time points (1: 14dpi; 2: 42dpi). The numbers indicate the groups according to Table 1 and the necropsy time point (group.necropsy time point). The most severe lesions are seen in the non-vaccinated groups (3 and 4) with severe interstitial pneumonia and massive alveolar necrosis. The lesions are milder and no longer statistically significant between the groups at the second necropsy time point. Histologic scores of the five lesions (pneumocytic hypertrophy and hyperplasia, septal infiltration with mononuclear cells, intra-alveolar necrotic debris, intra-alveolar inflammatory cell accumulation, and perivascular inflammatory cell accumulation for severity (0: no lesion; 1: mild; 2: moderate; or 3: severe) and extension (0: not present; 1: focal; 2: multifocal; 3: diffuse distribution) for the lung lobes) were: 1.1: 19/30; 2.1: 18/30; 3.1: 22/30; 4.1: 25/30; 5.1: 2/30; 1.2: 11/30; 2.2: 10/30; 3.2: 12/30; 4.2: 9/30; 5.2: 2/30.

**Figure 12 vaccines-10-00934-f012:**
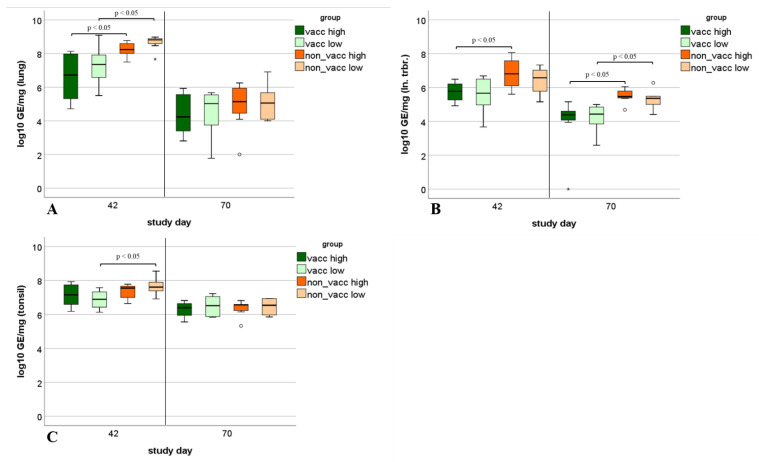
Viral load in tissue samples. Boxplots show 25th and 75th percentiles and median values, and whiskers show minimum and maximum values within 1.5 × interquartile range (IQR). Outliers are values with more than 1.5 × IQR (indicated using circles °), and extreme values are values with more than 3 × IQR (indicated using asterisks *). (**A**): Viral load in lung tissue (log_10_ GE/mg) from piglets necropsied on either study day 42 (first necropsy time point) or study day 70 (second necropsy time point) in each infected treatment group (vacc high: vaccinated, high dose [10^5^ TCID_50_ /dose] infected; vacc low: vaccinated, low dose [10^3^ TCID_50_/dose] infected; non-vacc high: non-vaccinated, high dose infected; non-vacc low: non-vaccinated, low dose infected). Significant differences (*p* < 0.05) between vaccinated and non-vaccinated groups are indicated. (**B**): Viral load in tracheobronchial lymph node (log_10_ GE/mg) from piglets necropsied on either study day 42 (first necropsy time point) or study day 70 (second necropsy time point) in each treatment group. Significant differences (*p* < 0.05) between vaccinated and non-vaccinated piglets are indicated. (**C**): Viral load in tonsils (log_10_ GE/mg) from piglets necropsied on either study day 42 (first necropsy time point) or study day 70 (second necropsy time point) in each treatment group. Significant differences (*p* < 0.05) between vaccinated and non-vaccinated piglets are indicated.

**Table 1 vaccines-10-00934-t001:** Experimental design: grouping of the animals.

Group	No. of Animals	Room	Treatment (Study Day 0)	PRRSV Challenge (Study Day 28)
1 (vacc ^1^ high)	16	A	Vaccination	10^5^ TCID_50_ ^2^/dose
2 (vacc low)	16	A	Vaccination	10^3^ TCID_50_/dose
3 (non-vacc high)	16	B	Sham treatment	10^5^ TCID_50_/dose
4 (non-vacc low)	16	B	Sham treatment	10^3^ TCID_50_/dose
5 (vacc control)	10	C	Vaccination	Sham inoculation

^1^ vacc: vaccinated; ^2^ TCID_50_: Tissue Culture Infectious Dose 50.

## Data Availability

All relevant data were included in the study. Raw data are available upon request by the authors.

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
