# Peer review of "Efficacy of a Modified Live Porcine Reproductive and Respiratory Syndrome Virus 1 (PRRSV-1) Vaccine against Experimental Infection with PRRSV AUT15-33 in Weaned Piglets"

_vaccines, 2022, doi:10.3390/vaccines10060934_

Round 1
Reviewer 1 Report
In this manuscript, the authors assessed the efficacy of the commercial modified live PRRSV-1 vaccine “Ingelvac PRRSFLEX® EU” in weaned piglets experimentally infected with PRRSV strain AUT15-33. This article gave some useful information about the experimental infection model of PRRSV with PRRSV AUT15-33 in weaned piglets and confirmed that this strain is a suitable challenge model both for the reproductive syndrome in sows and the respiratory disease in piglets.
In addition, the manuscript is well written, and logically presented. However, I would like to suggest that the authors consider the following comments.
- I would like to suggest the authors to provide more convincing evidence (eg. HE for histologic lung lesions or photos for macroscopic lung lesions) to better understand the efficacy of the commercial modified live PRRSV-1 vaccine “Ingelvac PRRSFLEX® EU” in weaned piglets experimentally infected with PRRSV strain AUT15-33.
- Please proofread the manuscript carefully. For example: in line 417, „Precision Livestock farming" should be "Precision Livestock farming".
Author Response
- I would like to suggest the authors to provide more convincing evidence (eg. HE for histologic lung lesions or photos for macroscopic lung lesions) to better understand the efficacy of the commercial modified live PRRSV-1 vaccine “Ingelvac PRRSFLEX® EU” in weaned piglets experimentally infected with PRRSV strain AUT15-33.
Response 1: Thanks for this suggestion. Histologic slides and a picture of the macroscopic lung lesions have been included.
- Please proofread the manuscript carefully. For example: in line 417, „Precision Livestock farming" should be "Precision Livestock farming".
Response 2: Thank you for the comment; it has been corrected in the manuscript.
Reviewer 2 Report
In this manuscript, the authors assessed the efficacy of the MLV vaccine “Ingelvac PRRSFLEX® EU” (Boehringer Ingelheim Vetmedica GmbH, Germany) against experimental infection with PRRSV AUT15-33 in weaned piglets. Animal challenge experiment was carried out and different doses of AUT15-33 were used as challenged viruses. Some representative data, including clinical signs, ADG, lung lesions and viral loads were analyzed and compared between vaccinated and non-vaccinated infected groups. The results indicated that piglets vaccinated with this commercial MLV vaccine shown reduced clinical signs, higher ADG and reduced viral shedding after infection with PRRSV AUT15-33. This study provides new data in evaluating the protective efficacy of commercial PRRSV vaccines against circulating strains. Overall, the paper is well written, with clear structures. The data presented also support the conclusions stated.
However, the followings points should be addressed:
- Line 75-76: corresponding references should be added here.
- For part 3.1, it would be better to show the rectal temperature data of different groups, as it is an important indicator of PRRSV infection.
- It would be better if the author could add some information about the survival rate of pigs infected with PRRSV AUT15-33 in this study.
- Discuss why the viral loads in serum and oral swabs of “vacc high group” was higher than that of “non-vacc high group” is interesting.
Author Response
- Line 75-76: corresponding references should be added here.
Response 1: A reference has been added (line 80). This statement is mainly based on the experience of the authors from routine diagnostic investigations (manuscript in preparation).
- For part 3.1, it would be better to show the rectal temperature data of different groups, as it is an important indicator of PRRSV infection.
Response 2: Thanks for the suggestion. Rectal temperature data of the different groups has been included (lines 242-255).
- It would be better if the author could add some information about the survival rate of pigs infected with PRRSV AUT15-33 in this study.
Response 3: No piglet died due to PRRSV AUT15-33 infection in the current study, that’s why we did not give any information about the survival rate of infected piglets. This information has been added (line 240).
- Discuss why the viral loads in serum and oral swabs of “vacc high group” was higher than that of “non-vacc high group” is interesting.
Response 4: Thanks for this suggestion. However, we do not really see that viral loads in oral swabs were higher in vacc high group compared to the non-vacc high group. In serum this was the case at individual time points; however, the differences were not statistically significant, and, in our opinion, levels of serum viral load were similar in all infected groups.
A native speaker has read the manuscript and checked it for English language and style.
Reviewer 3 Report
The paper by Duerlinger et al. deals with the efficacy of a commercially available vaccine against PRRSv infection, under experimental conditions. Although interesting, such paper raises the following concerns.
Line 49 – “symptoms” should be corrected as “clinical signs”.
Lines 64-82 – these sentences are useless at this point and/or should be included in the “materials and methods” paragraph.
Lines 83-88 – the aims of the paper should be better explained, remarking the added value of the present investigation. Line 83 sounds useless at this point.
Line 93 – data should be added about further respiratory pathogens (e.g., pigs were Mycoplasma hyopneumoniae free? PCV-2 positive or negative?). Moreover, pigs under investigation were confirmed as PRRS-free at their arrival?
Lines 96-98 – Authors should add more information about the following points:
- Why a real control group (i.e., neither infected nor vaccinated pigs) was not included in the present study?
- Why group 1-2 and 3-4 shared were placed in the same room? That prevented to evaluate the effect of the infectious dose.
Lines 106-110 – Authors should add more data about the main features of the inoculum (i.e., time of exposure and size of aerosol droplets). Moreover, they should explain why they used such infectious dose; did they follow a previous protocol?
Line 122 – please, add more details about clinical observations. Who carried out such observations? How long? Was he/she/they blind to the experimental conditions?
Line 128 – I understand the meaning of this sentence. However, it should be better explained (the term “second necropsy” sounds inaccurate).
Lines 134-140 – the inoculum was tested for other pathogens? Which inoculum was used as “sham treatment” in groups 3-4?
Lines 150-151 – how many samples were collected per each lobe? One? More? Please, add more data.
Line 156 – I do well understand the way to quantify the percentage of the lung affected by pneumonia. On the contrary, I cannot understand the quantification of the severity and location. In this respect, I noted that such data are not reported in the “results” paragraph. Please, add more details or delete.
Line 160 – how was scored the “extension” of lesions? As the percentage of each slide? Of the lung lobe? Please, add more details or delete. In addition, who carried out such scores? One or more operators? Was he/she/they blind to the experimental conditions? I consider important to better detail these points, which are somewhat subjective.
Lines 209-214 – “infected groups” means both vaccinated and not vaccinated? “vaccinated groups” means both infected and uninfected? Please, add more information. Line 212: “lower” means statistically significant or not? This paragraph appears inadequate.
Lines 215-223 – “more coughing”, “notable increase” etc. Once again, such differences are statistically significant or not?
Overall, considering the experimental design, it is impossible to properly evaluate the effect of the infectious dosage.
Line 224 – Please, add more information about the weight of piglets at T0. Was it homogeneous?
Lines 227-228 – p>0.05? Please, check and correct.
Lines 229-231 - Once again, such differences are statistically significant or not?
Line 235 – p<0.1? Please, check and correct.
Lines 235-238 - Once again, such differences are statistically significant or not?
Figure 3 – p<0.1? Please, check and correct. The entire caption should be checked and improved.
Lines 224-240 – these sentences are not clear and should be re-written.
Lines 248-257 - Once again, such differences are statistically significant or not?
Lines 264-281 - Once again, such differences are statistically significant or not?
Representative pictures should be provided about gross and microscopic lesions, thus showing the different “scores”. Moreover, data (even if negative) should be provided about other respiratory pathogens (viruses and bacteria).
Line 313 – p<0.1? Please, check and correct.
I consider it should be better to indicate each group with number (1-to-5) for greater clarity and to avoid repetition.
Throughout the entire investigation, it is not clear to me whether “low dose” and “high dose” (both vaccinated and not vaccinated) were compared or not. Please, add more details and results.
In my opinion, the “discussion” paragraph is too long and uselessly repeating results. I suggest to re-write this paragraph, dealing with the real discussion of results (strong and weak points) and highlighting any added value of the present study, when compared with the available literature.
Author Response
- Line 49 – “symptoms” should be corrected as “clinical signs”.
Response 1: Thank you for the comment; it has been corrected in the manuscript.
- Lines 64-82 – these sentences are useless at this point and/or should be included in the “materials and methods” paragraph.
Response 2: Thank you for the suggestion; it has been implemented in the manuscript.
- Lines 83-88 – the aims of the paper should be better explained, remarking the added value of the present investigation. Line 83 sounds useless at this point.
Response 3: Thank you for the suggestion, it has been implemented in the manuscript.
- Line 93 – data should be added about further respiratory pathogens (e.g., pigs were Mycoplasma hyopneumoniae free? PCV-2 positive or negative?). Moreover, pigs under investigation were confirmed as PRRS-free at their arrival?
Response 4: Thank you for the comment, the data have been added to the manuscript (lines 98-103).
- Why a real control group (i.e., neither infected nor vaccinated pigs) was not included in the present study?
Response 5: Since pivotal studies and our own experience out of other infection trials showed that the vaccine has no negative effects on animal health, we decided not to include another group (non-vaccinated, non-infected group) in the current study. Consequently, the number of animals used in this study could be reduced to a minimum.
- Why group 1-2 and 3-4 shared were placed in the same room? That prevented to evaluate the effect of the infectious dose.
Response 6: Of course, you are right, but unfortunately, it was logistically not possible to house the pigs in a different way. The piglets of the groups housed in the same air space did not have direct contact to piglets of the respective other group. However, vaccinated and non-vaccinated pigs were housed in different air space. The housing was one reason why we only compared vaccinated, low dose infected piglets with non-vaccinated, low dose infected piglets and vaccinated, high dose infected piglets with non-vaccinated, high dose infected piglets statistically.
The initial difference and subsequent accordance of the viral load between low dose and high dose infected animals suggests that the infection with different doses worked out despite the suboptimal housing conditions. Both, the lower and the higher infection dose led to symptoms and lung lesions in piglets, due to fast replication of the virus in piglets.
- Lines 106-110 – Authors should add more data about the main features of the inoculum (i.e., time of exposure and size of aerosol droplets). Moreover, they should explain why they used such infectious dose; did they follow a previous protocol?
Response 7: More precise information on the inoculum was provided (lines 120, 122-123).
The dose of 1x 105 TCID50 was chosen based on experience of the authors from previous studies. The infectious dose of 1x 103 TCID50 was additionally chosen to assess if PRRSV strain AUT15-33 is capable to cause clinical symptoms and lesions even at lower infectious doses.
- Line 122 – please, add more details about clinical observations. Who carried out such observations? How long? Was he/she/they blind to the experimental conditions?
Response 8: More information was added in the manuscript (lines 137-139).
- Line 128 – I understand the meaning of this sentence. However, it should be better explained (the term “second necropsy” sounds inaccurate).
Response 9: The sentence was changed as recommended (line 145).
- Lines 134-140 – the inoculum was tested for other pathogens? Which inoculum was used as “sham treatment” in groups 3-4?
Response 10: The inoculate itself was not tested for other pathogens. More information regarding the sham treatment of group 3 and 4 and the sham inoculation of group 5 has been added to the manuscript (lines 116-117, 125-126).
- Lines 150-151 – how many samples were collected per each lobe? One? More? Please, add more data.
Response 11: More information was added to the manuscript (lines 172-173).
- Line 156 – I do well understand the way to quantify the percentage of the lung affected by pneumonia. On the contrary, I cannot understand the quantification of the severity and location. In this respect, I noted that such data are not reported in the “results” paragraph. Please, add more details or delete.
Response 12: Thank you very much for this comment. You are completely right; there was a misleading sentence in Material and Methods, which we changed for clarification (line 179).
- Line 160 – how was scored the “extension” of lesions? As the percentage of each slide? Of the lung lobe? Please, add more details or delete. In addition, who carried out such scores? One or more operators? Was he/she/they blind to the experimental conditions? I consider important to better detail these points, which are somewhat subjective.
Response 13: More details on the evaluation of histologic lesions were added to the manuscript (lines 183-194).
- Lines 209-214 – “infected groups” means both vaccinated and not vaccinated? “vaccinated groups” means both infected and uninfected? Please, add more information. Line 212: “lower” means statistically significant or not? This paragraph appears inadequate.
Response 14: Thank you very much for this comment, the paragraph has been rewritten in the manuscript (lines 240-248).
- Lines 215-223 – “more coughing”, “notable increase” etc. Once again, such differences are statistically significant or not?
Thank you very much for this comment, the sentence has been rewritten in the manuscript (lines 259-262).
- Overall, considering the experimental design, it is impossible to properly evaluate the effect of the infectious dosage.
Response 16: I can understand your concerns about this, however, for this reason, we compared the results of vaccinated, low dose infected piglets to non-vaccinated, low dose infected piglets and vaccinated, high dose infected piglets to non-vaccinated, high dose infected piglets statistically. No comparison of high-dose versus low-dose infected piglets was performed in the statistical analysis. We checked the presented results in the manuscript to be precise and only describe the comparison between those groups but not between high dose and low dose infected piglets.
- Line 224 – Please, add more information about the weight of piglets at T0. Was it homogeneous?
Response 17: Thank you for this comment. More information about the weight of the piglets at study day 0 has been added (lines 268-270).
- Lines 227-228 – p>0.05? Please, check and correct.
Response 18: P-values < 0.05 are considered as statistically significant, p- values > 0.05 are considered as not statistically significant; p-values <0.1 were reported as numerical differences. This was also clarified in the methods (lines 236-237).
- Lines 229-231 - Once again, such differences are statistically significant or not?
Response 19: Thank you for the comment, the sentence has been removed and the paragraph has been revised in the manuscript.
- Line 235 – p<0.1? Please, check and correct.
Response 20: Thank you for the comment, it was added in material and methods, that p-values < 0.1 were considered as numerical differences.
- Lines 235-238 - Once again, such differences are statistically significant or not?
Response 21: The paragraph has been revised in the manuscript.
- Figure 3 – p<0.1? Please, check and correct. The entire caption should be checked and improved.
Response 22: The paragraph has been revised in the manuscript. It has been added in material and methods, that p-values < 0.1 were considered as numerical differences.
- Lines 224-240 – these sentences are not clear and should be re-written.
Response 23: Thank you very much for this comment, the paragraph was revised in the manuscript.
- Lines 248-257 - Once again, such differences are statistically significant or not?
Response 24: Thank you for this comment. In case of ELISA data, no statistics were calculated since the test is not validated to provide quantitative results but only to give a yes or no answer.
- Lines 264-281 - Once again, such differences are statistically significant or not?
Response 25: As stated in lines 321-322, no significant differences were detected between the infected groups.
- Representative pictures should be provided about gross and microscopic lesions, thus showing the different “scores”. Moreover, data (even if negative) should be provided about other respiratory pathogens (viruses and bacteria).
Response 26: Representative pictures were included as suggested by the reviewers. Since the involvement of other primary pathogens in the development of the lesions was excluded based on the health status of the piglets (see lines 98-103), no further investigations were performed.
- Line 313 – p<0.1? Please, check and correct.
Response 27: It has been clarified in the methods (lines 236-237) that p-values < 0.05 were considered as statistically significant; p-values < 0.1 were reported as numerical differences in the manuscript.
- Throughout the entire investigation, it is not clear to me whether “low dose” and “high dose” (both vaccinated and not vaccinated) were compared or not. Please, add more details and results.
Response 28: The main objective was to compare vaccinated with non-vaccinated piglets either infected with high dose or low dose of PRRSV in order to evaluate vaccine efficacy. Therefore, results of vaccinated, low dose infected piglets were compared statistically to non-vaccinated, low dose infected piglets and vaccinated, high dose infected piglets were compared to non-vaccinated, high dose infected piglets. Since low dose and high dose infected piglets were housed in the same air space, no comparisons were made between those groups. We double checked the manuscript to make sure only the comparisons which were done based on statistical methods between vaccinated and non vaccinated piglets infected with the same virus dose are reported.
- In my opinion, the “discussion” paragraph is too long and uselessly repeating results. I suggest to re-write this paragraph, dealing with the real discussion of results (strong and weak points) and highlighting any added value of the present study, when compared with the available literature.
Response 29: Thank you for this constructive criticism. Some paragraphs in the discussion in which results were repeated have been deleted and more discussion on the weakness of the study has been included.
A native speaker has read the manuscript and checked it for English language and style.
Round 2
Reviewer 3 Report
Overall, the manuscript appears much improved in the “Introduction”, “Materials & Methods” and “Results” paragraphs, although some issues remained unsolved. Few further comments:
- Introduction can be further shortened.
- Figure 8 is of poor quality and does not provide useful information to the readers. If this is the best one and you don’t have good pictures of the different groups, I suggest deleting it.
- Figure 12: microscopic lesions are evident and of good quality. Please, provide the score assigned to each picture within the figure’s caption.
I still have some concerns about the discussion:
- First of all, it appears unbalanced (too long and redundant).
- Lines 460-473, as well as lines 486-496, seem “out of focus”.
- Lines 572-580 sound highly speculative and not supported by data.
I suggest to follow a "point by point" approach, dealing with the aims of the study (starting with the development of an effective challenge protocol…), briefly reporting results and discussing them, highlighting the “new” contributions resulting from this investigation.
Author Response
Response to Reviewer 3 Comments
- Introduction can be further shortened.
Response 1: Thank you for the comment, the introduction was further shortened (lines 36-41).
- Figure 8 is of poor quality and does not provide useful information to the readers. If this is the best one and you don’t have good pictures of the different groups, I suggest deleting it.
Response 2: Thank you for the suggestion; since there is no comparable picture of higher quality available, figure 8 has been deleted.
- Figure 12: microscopic lesions are evident and of good quality. Please, provide the score assigned to each picture within the figure’s caption.
Response 3: Thank you for the suggestion, the scores assigned to the histological pictures have been implemented in the manuscript (lines 389-395).
- First of all, it appears unbalanced (too long and redundant).
Response 4: Thank you for the constructive criticism. The discussion has been further shortened.
- Lines 460-473, as well as lines 486-496, seem “out of focus”.
Response 5: Thank you for the comment. The respective paragraphs have been shortened (lines 428-437 and lines 455-461).
- Lines 572-580 sound highly speculative and not supported by data.
Response 6: Thank you for this comment. I am aware that this statement sounds speculative (that’s why I tried to phrase it carefully), but in our opinion, it is supported by the data (viremia and histology) of the current study.
A native speaker has read the manuscript and checked it for English language and style.